# Anal Cancer in High-Risk Women: The Lost Tribe

**DOI:** 10.3390/cancers15010060

**Published:** 2022-12-22

**Authors:** Micol Lupi, Danielle Brogden, Ann-Marie Howell, Paris Tekkis, Sarah Mills, Christos Kontovounisios

**Affiliations:** 1Department of Surgery and Cancer, South Kensington Campus, Imperial College London, London SW7 2AZ, UK; 2Department of Colorectal Surgery, Chelsea and Westminster Hospital NHS Foundation Trust, 369 Fulham Road, London SW10 9NH, UK

**Keywords:** HSIL, SCC, HPV, genital, anal, women

## Abstract

**Simple Summary:**

Anal cancer rates are on the rise, especially for women. Women with a background of genital dysplasia or cancer have been shown to be at a higher risk of anal cancer than the general population. Despite this, very little has been done to educate women of the increased risk and as a result women tend to present later with advanced disease. As anal cancer has a known precancerous precursor, there is potential for patients to receive preventative treatment prior to anal cancer development. Although, due to the rarity of the disease, there are currently no clinical guidelines for the prevention of anal cancer in women. There are likely to be missed opportunities in preventing anal cancer progression from treatable dysplastic precursor lesions. This review presents the current evidence supporting the screening, treatment, and surveillance of anal precancerous lesions in women with genital dysplasia or cancer.

**Abstract:**

In developed countries the incidence of anal squamous cell carcinoma (SCC) has been rising; especially in women over the age of 60 years who present with more advanced disease stage than men. Historically, anal SCC screening has focused on people living with Human Immunodeficiency Virus (HIV) (PLWH) who are considered to be at the highest risk of anal SCC, and its precancerous lesion, anal squamous intraepithelial lesion (SIL). Despite this, women with vulval high-grade squamous epithelial lesions (HSIL) and SCCs have been shown to be as affected by anal HSIL and SCC as some PLWH. Nevertheless, there are no guidelines for the management of anal HSIL in this patient group. The ANCHOR trial demonstrated that treating anal HSIL significantly reduces the risk of anal SCC in PLWH, there is therefore an unmet requirement to clarify whether the screening and treatment of HSIL in women with a prior genital HSIL is also beneficial. This review presents the current evidence supporting the screening, treatment, and surveillance of anal HSIL in high-risk women with a previous history of genital HSIL and/or SCC.

## 1. Introduction

Anal squamous cell carcinoma (SCC) is a relatively rare pathology affecting 1–2 per 100,000 people [1,2]. However, in developed countries it is increasing in incidence; a study looking at National Cancer Outcomes and Service Data (COSD) in England, reported a rise of 23.4% between 2013 and 2017 [3]. People living with HIV (PLWH), in particular men who have sex with men (MSM), are the group of patients considered at highest risk of anal SCC and its precancerous lesion anal squamous intraepithelial lesion (SIL) [4]. As a result, most research and, consequently, clinical practice has revolved around anal SCC surveillance and prevention in this patient group [5,6,7,8]. 

Anal cancer incidence is however highest in women compared to men [2,3,9,10], 2.52 versus 1.26 per 100,000 people, respectively [3], and is rising more rapidly in women compared to men [2,10], 28.6% versus 13.5%, respectively, in England between 2013 and 2017 [3]. More importantly, women have been shown to be more likely to present late with advanced cancers compared to men [3,9]; a study looking at Surveillance, Epidemiology and End Results Program (SEER) data found women more likely to present with advanced staging (*p* < 0.001), to receive radiotherapy (2.67 times, 95% CI 2.55–2.81, *p* < 0.001) and die of anal cancer than men (1.25 times, 95% CI 1.17–1.32, *p* < 0.001) [9]. Similarly, another study using COSD saw an increase in women presenting late with metastatic cancers, with 45.6% of women presenting with either Stage 3 or 4 disease [3].This is important when considering survival rates which are significantly better for stage 1 and 2 compared to stage 3 and 4 disease; 90% and 80% 5 year survival rates versus of 60% 5 year and 55% 1 year survival rates, respectively [11]. Of note, recent work has shown women with vulval high-grade squamous epithelial lesions (HSIL) and squamous cell carcinomas to have similar relative risks of anal HSIL and SCC as some PLWH [4]; HSIL and SCC of the cervix and vagina are also significant risk factors for anal HSIL and SCC [4].

Despite this there are no guidelines for the management of anal HSIL in high-risk women. Lack of awareness regarding the concomitant risk is a potential issue. The ANCHOR trial demonstrates that treating anal HSIL in PLWH significantly reduces the risk of anal SCC [12,13]. There is a need to identify all populations at high-risk of anal HSIL and SCC and develop clinical pathways for early identification and treatment. This review summarises the current evidence supporting the screening, treatment, and surveillance of anal HSIL in high-risk women with a previous history of genital HSIL or SCC. 

## 2. High-Risk Human Papillomavirus Pathogenesis

Around 4.8% of all cancers worldwide are attributed to high-risk Human Papillomavirus (hrHPV) infection [14]. Human Papillomavirus is a non-enveloped double-stranded DNA virus with over 100 known genotypes, approximately 30 genotypes which are sexually transmissible and several genotypes have been shown to be carcinogenic for the anogenital region; typically hrHPV type 16 and 18 [15,16]. It is thought that almost all sexually active women and men will be infected with HPV in their lifetime, regardless of the their sexual practices; with up to 50% of hrHPV infections clearing within 6 months to 2 years after acquisition and under 10% of infections persisting eventually causing dysplasia [17]. hrHPV is responsible for 40% of vulval cancers, 70% of vaginal cancers, 100% of cervical cancers and 90% of anal cancers [14,18]; with HPV16 being the most carcinogenic HPV genotype in the anus [19].

HPV interacts with squamous epithelial cells via viral proliferation within transient lesions, but can cause persistent infection and eventual progression to pre-cancerous lesions [16]. Progression to high-grade lesions is related to the deregulation of oncoproteins E6 and E7 which affect the host’s cell cycle by promoting the degradation of p53 and inhibiting Retinoblastoma protein (pRb), respectively [16,20]. 

pRb is a cell cycle protein involved in the maintenance of the Restriction Point, the check point preventing cell cycle progression, from the G1 to the S phase [21], (see Figure 1). To allow cell cycle progression pRb, which is at this point hypo-phosphorylated and bound in a complex with E2F (a transcription factor), is phosphorylated by the cyclin D/cdk4/6 complex allowing the release of E2F which when activated promotes the transcription of genes required for S phase progression and mitosis [21]. 

P53 has complex interactions with the cell cycle and plays a key role in regulating cell death. Among other pathways, p53 activates p21 a cyclin-dependent kinase inhibitor which regulates the G1/S checkpoint, as well as activates apoptotic genes and pathways [22]. Thus, oncoproteins E6 and E7 dysregulate the cell cycle and promote neoplastic change by increasing cell proliferation in the basal layers and impairing the host cell from repairing mutations. 

The HPV viral genome contains other genes (E1, E2, E4, E5, E6, E7, L1 and L2) which code for proteins involved in multiple actions. These include binding of viral particles to the basement membrane of the host cell, permitting the transport of virions to the nucleus of the host cell, contributing to viral integration, regulating host cell anti-apoptotic pathways, regulating transmission of viral particles, activating cell proliferation, and impairing host immune response [23,24]. Integration of the viral HPV16 DNA into the host genome has been shown to be one of the key steps for the progression of premalignant cervical lesions [25]; with increasing viral load and markers of DNA integration such as an E2/E6 ratio of 0 or greater than 0 and less than 0.8 being associated with increasing lesion severity [26]. Integration frequently leads to the loss of expression of E2, the viral transcriptional repression, consequently leading to the unregulated expression of E6 and E7 and inactivation of p53 and pRB, respectively [20]. This causes cell proliferation, genomic instability and eventual malignant transformation [16]. Nevertheless, viral integration, at least within the E2 target, is not seen in all cervical cancers, with one study reporting episomal (not integrated) HPV16 DNA in 18% of cases, mixed integration (presence of both integrated and episomal DNA) in 45% of cases and full integration in 37% of cases [26]. Research looking into HPV16 DNA integration in anal cancer is unfortunately more limited. Valmary-Degano et al. [20] carried out a comprehensive study and found HPV16 integration in 77.8% (42/54) of cases. However, this included both full and mixed integration; 70.4% of cases harbored both episomal and integrated HPV16 and full integration was observed in only 7.4% of cases. Similar integration rates were seen for anal HSIL in HIV positive men [27].

## 3. Anal Squamous Intraepithelial Lesions and Their Natural History

Anal squamous intraepithelial lesions are the neoplastic precursors of anal squamous cell carcinoma. The classification of all anogenital neoplastic lesions was recently standardised in 2012 as a result of the Lower Anogenital Squamous Terminology (LAST) project [28], which aimed to align diagnostic terminology with current knowledge of HPV biology, increase reproducibility, establish one accepted nomenclature classification and improve one’s ability to predict a patient’s true cancer risk. The classification entails the division of squamous intraepithelial lesions into high or low grade squamous intraepithelial lesions (HSIL or LSIL, respectively); with LSIL encompassing the previously known low grade anal intraepithelial lesions (AIN 1) or low grade anal intraepithelial neoplasia (LGAIN) and HSIL the moderate and high grade anal intraepithelial lesions (AIN 2 and 3) or high grade anal intraepithelial neoplasia (HGAIN). In addition to this, the LAST guidelines also recommend the use of p16 staining as a biomarker to distinguish HSIL from LSIL in cases where there is diagnostic uncertainty [28]. 

Not all anal HSIL progresses to SCC with some cases spontaneously regressing to LSIL. There is little consensus on the progression and regression times between anal LSIL, HSIL and SCC. Lee et al. [29] and Fuchs et al. [30] quote a time period of around 3 years for HSIL to progress to SCC. Schofield et al. [31] found 9% of anal HSIL patients progressed to SCC over a median of 63 months (range 14–120 months) and Watson et al. [32] found this in 13% of anal HSIL patients over a median of 60 months (range 80–112). Others have reported a 9.5% 5-year incidence of anal SCC after HSIL, or 1.9% per year [29]. It is however worth noting that these figures are influenced by factors such as treatment and host immunity and are therefore not absolute. In the same study, Schofield et al. [31] found immunocompromised or patients with multifocal disease to be at the highest risk of progression to invasive cancer; with 50% of systemically immunocompromised patients with anal HSIL progressing to invasive disease within 5 years compared to no progression to invasive disease in the immunocompetent group. The ANCHOR study published the most recent available data with respect to anal HSIL progression to cancer and reports rates of progression of 402 per 100,000 person years (95% confidence interval (CI) 262–616), a rate which was lower for those in the treatment arm (173 per 100,000 person years (95% CI 90–332) and also influenced by the size of lesion (1047 vs. 185 per 100,000 person years in those with a lesion more than or less than 50% of the anal or perianal area, respectively) [12,13]. Older age, larger lesion size and persistence of HPV infection are all accepted predictors of progression and will also contribute to the observed variability in published progression rates [33,34,35]. As for regression, one study found 23.8% of patients to spontaneously regress from HSIL to LSIL or negative biopsies over 102.2 person years follow-up [35] and another reported clearance rates of 22.2 per 100 person years [34]. Most these studies were again carried out in men living with HIV and not HIV negative high-risk women and therefore may not reflect the natural history of the pathology in women, with that said women with a background of genital HSIL with anal HSIL tend to be older, are less likely to successfully clear HPV, have multifocal disease and present late with large lesions which are higher risk for progression to SCC. 

There is now research investigating biomarkers predicting which anal HSIL lesions are most likely to progress or regress. Host cell DNA methylation markers have shown to be promising with respect to clinical decision-making in cervical HSIL and cancer. For example, DNA methylation markers, such as FAM19A4/miR124-2 methylation have been investigated and validated for application in cervical screening [36,37]; with a methylation negative test representing a significantly lower 14 year cervical cancer risk among HPV- positive women compared to a negative cytology test [38].

Similar work is being carried out in anal HSIL with a methylation marker panel containing ASCL1 and ZNF582 being used to detect anal cancer and HSIL. Van der Zee et al. [39], found higher methylation levels in HSIL to be associated with a higher risk to cancer progression. The next study, of which results are still not published, will be looking at HSIL methylation predicting regression [33]. Identification of biomarkers determining which individuals are most likely to regress or progress will be invaluable in risk stratification and in the identification of patients most likely to benefit from treatment and closer surveillance. 

## 4. Risk Factors for Anal hrHPV Colonisation in Women

hrHPV is transmitted via physical contact, this is more often than not via sexual intercourse, both penetrative and non-penetrative, between two individuals. The number of sexual partners (>5 [40]), has been shown to increase the risk of developing anal HSIL in women [41], by increasing the frequency of hrHPV exposure, which becomes increasingly more difficult to clear over the age of 30 years [42]. 

Similarly, concurrent infection with genital chlamydia [43], syphilis, genital herpes simples virus and gonorrhoea [44] have also been shown to amplify the risk of anal HSIL. The mechanism behind this is still not fully understood; McCloskey et al. [44] argue that this could be more of a reflection of sexual behaviour, however the other theory revolves around whether the concurrent infection increases the risk of hrHPV transmission by making mucous membranes more susceptible to infection [44,45].

Interestingly, whilst receptive anal intercourse appears to be an important risk factor for anal hrHPV infection in men [46], in women it does not appear to be necessary for anal infection. Some studies have identified a statistically significant relationship between anal HSIL and HPV infection and anal intercourse [40,41], however as shown in the study by Slama et al. [40], concurrent anogenital infection was strongly associated with any type of sexual contact with the anus, including non-penetrative anal sex (OR 2.62, *p* = 0.008) [40]. In concordance with this, a recent study looking at type-specific concurrent anogenital HPV infection among young women and MSM, found that type-specific concurrent anogenital HPV detection was common among women but not MSM, moreover this was not associated with receptive anal intercourse for women [43]. This suggests that concurrent colonisation may be from sex without penile-anal penetration or from auto-inoculation between the genital and anal region, or vice versa [43]. 

Regardless of transmission method there is a relationship between genital and anal HPV infection. A recent metanalysis identified anal HPV16 prevalence of 41% (447/1097) in cervical HPV16 positive patients versus 2% (214/8663) in cervical HPV16 negative women [47]. Pooled analysis in this study also showed a significant association at the HPV type-specific level between the anus and cervix [47]. 

The authors of this review therefore suggest that anal intercourse is less important than the number of sexual partners for women, with anal HPV exposure likely arising from sexual contact [48]. 

## 5. Risk Factors for Chronicity of hrHPV Infection in Women

Whilst we might understand the risk factors for hrHPV colonisation, the reasons why some individuals develop chronic HPV infection are still not fully understood and require further investigation. Some well-established risks for HPV persistence are listed below but by no means fully explain the aetiology.

### 5.1. Smoking

Smoking is a well-established risk factor for anal cancer in both sexes [45,49]. In women, smoking has been shown to be a particularly strong risk factor for multizonal anogenital disease; Slama et al. report a smoking history of more than >60 cigarettes per week to be a significant (OR, *p* = 0048) risk factor in women with multiple concurrent anogenital HPV infections [40]. Albuquerque et al. did not find an association with the number of cigarettes smoked but found active smoking, as opposed to previous smoking, to be a significant risk factor [50].

### 5.2. Immunosuppression

Immunosuppression secondary to chronic diseases, transplantation and associated medications have also been shown to correlate to chronic anogenital hrHPV infection and anogenital HSIL and or cancer [50]. A comprehensive, systematic review reported anal cancer incidence rates of 13, 10, 6, 3 per 100,000 people for transplant recipients, Systemic Lupus Erythematosus patients, Ulcerative Colitis and Crohn’s disease patients, respectively [4]. This is compared to the baseline incidence of 1–2 per 100,000 people [1,2]. 

More specifically to women, a register-based cohort study in Denmark, Reinholdt et al. found a positive relationship between diabetes and HPV related anogenital HSIL and SCC [51]. 

The same group also looked at female renal transplant patients and found 10–15% and 5–12% of female transplant patients below the age of 40 develop cervical or vaginal/vulva/anal HSIL, respectively, within 20 years of transplantation. This was compared to 4–8% and 0.2–0.4%, respectively, in non-transplant patients [52]. 

Inflammatory bowel disease (IBD) patients have also been shown to be at increased risk of developing anal cancer. In a study of 61,648 patients, out of which 837 had IBD, Segal et al. found IBD patients to have significantly higher age-standardised rates (ASR) of anal cancers than the non-IBD population, 5.5 vs. 1.8 per 100,000, respectively [53]. Their results did not compare rates between men and women; however, they did find significantly higher rates of cervical cancer in the IBD group (5.2 vs. 4.6 per 100,000, respectively, *p* = 0.042) [53], making IBD a potential risk factor for all anogenital neoplasia and SCC. 

### 5.3. Human Immunodeficiency Virus

Human immunodeficiency virus is the most researched risk factor for chronic hrHPV infection, anogenital HSIL and SCC in the literature. Clifford et al. carried out an extensive systematic review looking at anal SCC incidence rates in high-risk populations and in the PLWH populations found rates of 85, 32 and 22 per 100,000 in MSM, non-MSM and females, respectively [4]. There is no doubt that HIV infection increases the risk of anogenital neoplasia and subsequent SCC, likely due to its immunosuppressive effects that permit persistent hrHPV infection [54]. 

With that said, for women with genital hrHPV, this risk appears to be independent of HIV status. In a recent systematic review looking at the cervical determinants of anal HSIL, anal HSIL was associated with cervical hrHPV in both HIV negative women (24%, 33/138, *p* < 0·0001) and HIV positive women (17%, 31/186, *p* < 0·0001) [47]. Similarly anal HSIL was associated with cervical histopathology in HIV negative women (up to 22% (59/273) in cervical HSIL, *p* < 0·0001) as well as HIV positive women (up to 25% (25/101) in cervical HSIL, *p* < 0·0001) [47]. It is also worth noting that Clifford et al. [4] reports incidence rates of anal cancer in women with vulval cancer and vulval precancer which are higher than that for women living with HIV (48 and 42 vs. 22 per 100,000 people, respectively). 

## 6. The Perineum, a Reservoir for Anogenital hrHPV in Women

The evidence now indicates that a background of persistent genital hrHPV infection, or the resultant genital HSIL and/or SCC, is an independent risk factor for anal HSIL and SCC in women. The perineum is thought to act as a reservoir for hrHPV which can easily spread and develop HSIL anywhere in the anogenital region [3]. Given that hrHPV exposure is most common in sexually active women under 30 years of age [42,55] and that cervical HSIL and cancer tends to present in women under the age of 50 [55], whilst vulval and anal cancer in those above the age of 50 years [2,56], a hypothesis is that the cervix acts as the initial reservoir of hrHPV which over time spreads over to gradually more distant areas. Persistent cervical hrHPV infection, HSIL and/or SCC may therefore act as a potential early determinant of latent vulval and/or anal disease. 

There is evidence supporting this theory. Papatla et al. [57] found patients diagnosed with cervical cancer between the ages of 20 and 53 years to have an increased risk of developing anal cancer (SIR 3.53, 95% CI 1.15–8.23), a risk which was significant 10 or more years after the cervical cancer diagnosis and not seen in those patients diagnosed with cervical cancer after the age of 53 years. Similarly Kalliala et al. [58], found the relative risk of vaginal (10.84, CI 5.58–21.10, *p* < 0.001), vulval (3.34, CI 2.39–4.67, *p* < 0.001) and anal cancer (5.11, CI 2.73–9.55, *p* < 0.001) to be higher in women previously treated for CIN compared to the general population. Another study also found a significant relationship between cervical cancer and subsequent anal cancer with a median time lag between the two pathologies of 20 years [59]. 

With that said, Clifford et al. [4] carried out an extensive systematic review examining anal SCC incidence rates in high-risk populations and amongst women with genital hrHPV associated pathology reported incidence rates of 9, 6, 48, 42, 10, 19 per 100,000 people/year for women with cervical cancer, cervical precancerous lesions, vulval cancer, vulval precancerous lesions, vaginal cancer and vaginal precancerous lesions, respectively (compared to the baseline incidence of 1–2 per 100,000 people/year [1,2]). This suggests that the anatomical proximity of previous genital HSIL may be a predictor of anal HSIL development, with anal cancer incidence being higher in patients with vulval, followed by vaginal and cervical disease. This has been replicated in numerous other studies; one which found rates of abnormal anal cytology to be higher in women with vulval than cervical disease, 47.1% versus 12.5%, respectively [60]; another which found anal HSIL in 67% and 13% of their vulval and cervical dysplasia cohorts, respectively [61]. On the other hand, other studies have found stronger relationships between cervical HSIL or SCC and anal SCC (SIR 5.9 and 6.3, respectively) versus vulval SCC (SIR 4.4 and 1.9, respectively) [62] and also report cervical SCC occurrences after diagnosis of anal and vaginal cancer [62]. It worth noting that the incidence rates of anal cancer in these patient groups are also affected by age, especially with cervical HSIL which affects younger women who are less likely to have a synchronous anal cancer, than women with vulval pathology who are older; Clifford et al. categorised incidence by age group and found incidence rates of 1.3, 8.1 and 15 per 100,000 people/year for women under 40, between 40–59 and over the age of 60 years [4].

With a 10 to 20-year lag between cervical and anal disease, limited research, lack of anal disease surveillance and the fact that cervical screening has only recently changed from cytology to hrHPV testing, it is possible that women affected by vulval disease were colonised with hrHPV in the cervix earlier in life. A study looking at women with multizonal anogenital disease found the cervix to be the most affected initial site of anogenital disease. Perianal and anal canal pathologies on the other hand, were most commonly seen as secondary new sites of disease later in life [50].

Unfortunately, there is heterogeneity in these studies especially with respect to age, grade of SIL and methodology. Studies including cervical pathology look at a younger patient cohort than studies focusing solely on vulval pathology, thereby potentially underestimating the prevalence rate for anal HSIL or SCC. The grade of the genital SIL varies between studies, with studies looking at both HSIL and LSIL potentially underestimating the prevalence rate for anal HSIL or SCC. The method of anal HSIL screening can be inadequate, with studies not using high resolution anoscopy (HRA), the gold standard for anal HSIL detection [63], potentially underestimating the prevalence of anal HSIL or SCC. These inconsistencies will all have an impact on how well we can establish the true risk and incidence of anal HSIL or SCC in patients with different genital lesions. 

The evidence nevertheless suggests that the incidence of anal hrHPV or SIL or SCC is higher in women with genital hrHPV or SIL or SCC [17,59,60,61,64,65]. A risk, which has been shown to increase with increasing number of anogenital SIL foci; Schofield et al. [66] found 19% of patients with cervical HSIL to also have evidence of anal SIL, in comparison to 57% of women with more than 1 focus of genital SIL. Similarly, Albuquerque et al. [50] found 98% of their patients with multizonal disease to have a previous history of anogenital HSIL or SCC in one of the anogenital zones.

## 7. Understanding Anal Cancer Trends in Women

Anal SCC incidence is on the rise especially for women [2,3] and, as previously mentioned, women are presenting late with advanced anal cancers [3,9].The sexual revolution of the 1960s and 70s played an important role in promoting more risky sexual behaviour, earlier age at first intercourse and a higher number of sexual partners [2,67,68]. This has been associated with an increased prevalence of all sexually transmissible infections including hrHPV and consequently its associated anogenital dysplasia [2,67,68]. However, whilst, thanks to the cervical screening programme, cervical cancer incidence and mortality has been significantly declining since the mid-20th century [10,69,70], that for other hrHPV related pathologies which present later on in life and have no official screening service has been on the rise. This is observed with anal and vulval cancer [10,70,71].

Education is another important factor. Due to lack of research and anal SCC prevention strategies in this patient group, women with genital hrHPV driven pathology are simply not aware of their increased risk of developing further synchronous or metachronous anogenital dysplasia. Instead, with the cervical screening programme ending at 65 years of age, women at significant risk of anal cancer are left with no surveillance nor education at the time they are most likely to develop anal pathology. Even most cervical and vulval cancer guidelines fail to include digital anal examination as part of routine follow-up [72,73,74,75,76,77]. An oversight given that the incidence rates for vaginal and vulval cancers have been shown to fall over successive years after a diagnosis of cervical HSIL [78]; thereby demonstrating the potential benefit of also including anal inspection in routine follow-up. 

Diagnostic uncertainty is an issue. Anal HSIL and SCC lesions often present with relatively non-specific symptoms such as perianal discomfort, a palpable lump, itching and bleeding. Many lesions are flat in nature, as opposed to the warty appearances of LSIL [79], and can be relatively asymptomatic. This can create diagnostic ambiguity for general practitioners who may not have the experience to differentiate anal HSIL and SCC from more common pathology such as haemorrhoids. This could ultimately lead to incorrect management and delayed specialty referral and diagnosis. Complicating the matter is the fact that the self-reporting of anal symptoms is not necessarily associated with the presence of HSIL [79] and that between 3–10% of macroscopically normal perianal lesions in high-risk populations have been shown to contain SIL [80]. Understanding the risk profile for individual patients is therefore key, however a complete genital hrHPV history may not always be taken given that its relationship with anal pathology is not always common knowledge. In addition, given the 10 to 20 years gap between genital and anal pathology patients may not be able to remember their full genital history.

Like HIV and other sexually transmitted infections there can be a social stigma attached to HPV which can delay and prevent presentation when symptoms first arise, especially in an area as private and sensitive as the anogenital region. This has been shown to be particularly true for certain ethnic groups where cultural reasons also play an important role [81,82,83]. A study looking at COSD in England between 2013 and 2017 found that Black African or Black Caribbean women were more likely to be diagnosed with anal SCC at a younger age with more advanced staging and not receive treatment for their malignancy [3]. Similar trends have been reported for cervical cancer, with Black women having higher incidences of disease [84], higher cervical cancer mortality [85] and lower compliance with treatment once cervical dysplasia has been identified [86]. Ethnic minorities have been shown have a lower awareness of their cancer risk [82] and availability of screening programmes [87] which in turn translates to a delayed diagnosis and poorer prognosis. 

Socioeconomic deprivation is also relevant, with high deprivation being associated with difficulties in accessing healthcare due to lower education levels and cancer risk awareness [88] as well as unstable employment and housing affecting one’s ability to attend appointments and receive hospital correspondence [89]. Socioeconomic deprivation is known to impact the incidence and prognosis of HPV related cancers, in particular cervical cancer [84,90,91,92]. For anal cancer, there is also some evidence to suggest that low socioeconomic status is associated with advanced staging and lower survival rates [9] and that that ethnic minorities are more likely to be socially deprived [3].

There is still much to learn with respect to the underlying causes responsible for the currently observed rise in anal cancer in women and the associated poor prognosis. HPV vaccination is likely to eventually reduce the incidence of anal SCC in women over time, however the vaccination programme only started in 2008 and we are not likely to see its true impact on anal SCC before 2050 when vaccinated women reach the peak incidence age of anal SCC. Moreover, compliance with vaccination presents its own challenges with ethnicity, economic status, attitudes towards vaccination and education all having an impact [93,94]. Without any intervention targeting the secondary prevention of anal SCC in high-risk women, we are likely to continue to see a rise in incidence for some time. There is a clear need to improve education and services for this patient group as well as tackle barriers negatively impacting the interaction of certain high-risk women with such services.

## 8. Screening, Surveillance and Treatment of Anal HSIL in High-Risk Women: Is It Necessary?

Until recently, the evidence supporting the need for HSIL treatment in high-risk populations was lacking. Many argued whether it was in the patient’s best interest to undergo uncomfortable treatments to the anus, considering the relatively low progression rate to SCC, the fact that some lesions spontaneously regress, the lack of evidence demonstrating that treatment prevents progression to invasive disease and evidence demonstrating high rates of anal HSIL recurrence (>50%) after treatment [95]. With that said, despite the lack of high-quality evidence, many medical societies support anal HSIL surveillance and consideration of treatment, acknowledging the increased risk of anal cancer in certain populations and the clear benefits of early disease detection [6,7,8,96,97,98,99,100,101,102,103,104,105].

The ANCHOR trial [12,13] published in June 2022, was pivotal in changing the gold-standard of anal HSIL management by finally demonstrating the treatment of anal HSIL, mainly by electrocautery ablation (83.6%), to be effective in preventing its progression to SCC, at least in people living with HIV. Participants in close surveillance group were 57% more likely to progress to SCC than in the active treatment arm, with only 9/2071 cases of cancer progression in the active treatment arm vs. 22/2080 in the close surveillance arm. The study demonstrated that patients with anal HSIL should be offered treatment and that close surveillance alone, even with HRA, is not sufficient to prevent progression to SCC. Whether this applies to HIV negative women with genital HSIL or SCC, remains to be established, however the potential benefit of anal HSIL treatment must be acknowledged for all high-risk groups. These findings will hopefully drive research establishing which high-risk populations to screen, the best tools for screening, the screening and surveillance intervals and the most effective anal HSIL treatments. This will need to be established independently for different high-risk patient groups which will respond differently to different approaches. 

## 9. A Potential Screening, Treatment and Surveillance Model for High-Risk Women

There are currently no global standardised screening and treatment guidelines for anal HSIL. Different medical societies in different geographical regions have created their own clinical recommendations for clinical practice [6,7,8,96,97,98,99,100,101,102,103,104,105]. Nevertheless, like the ANCHOR trial [12,13], these mainly focus on PLWH, especially HIV positive men who have sex with men, and although they acknowledge genital HSIL and or SCC to be a risk factor, they underestimate the risk and do not give any guidance with respect to the surveillance of this patient group. 

The increased anal cancer risk in this patient group has however not gone unnoticed and societies such as the International Anal Neoplasia Society (IANS) do advocate for anal HSIL screening in this patient group [106]. In a recent online survey on provider preferences for anal cancer prevention screening completed by 140 IANS affiliates, 99% of respondents recommended routine anal cancer screening for patients with vulval cancer, in addition, HRA was recommended significantly more frequently for MSM living with HIV and patients with vulval cancer compared to others [106]. It is important to note that in the 1950s, in high-income settings, cervical cancer incidence rates of 30–40 per 100,000 were considered high enough to implement Papanicolaou smear screening in women over 35 years of age [4]; which in turn would qualify all women with vulval pathology to be considered for anal HSIL screening, with incidence rates of 42 and 48 per 100,000 for those with vulval HSIL and SCC, respectively [4]. 

With more research in this patient group, it would not be surprising if we eventually establish similar incidence rates for women with cervical pathology, given that those women with vulval pathology are also likely to have active or, at the very least, cleared cervical hrHPV infection. 

### 9.1. Screening

Multiple screening techniques for anal HSIL exist. These vary in cost, accessibility, sensitivity, and specificity. 

#### 9.1.1. The Digital Anorectal Examination

The digital anorectal examination (DARE) represents an inexpensive and essential technique in the assessment of the anus. IANS have standardised the technique to maximise its sensitivity and argue it is a useful tool for the detection of cancers 0.3 cm or greater in diameter [107]. It does not however detect HSILs or superficially invasive squamous carcinomas of the anus which require more sensitive tests. Moreover, detection rates will vary with clinician expertise and experience. 

#### 9.1.2. Anal Cytology and HPV Testing

Anal cytology or Papanicolaou test is a widely accepted and used technique for the screening of anal cancer, which samples cells at the squamocolumnar junction where most HSIL occurs [108], it is not appropriate for perianal disease given its hyperkeratosis, low cellularity and consequent low sensitivity [108]. 

A recent systematic review and metanalysis found anal cytology testing summary sensitivity and specificity estimates of 81% and 62% [109]. The sensitivity of anal cytology has been shown to be higher in immunosuppressed patients and in the presence of larger lesions [108,109]. A study comparing the performance of anal cytology vs. HRA and histology in women with anogenital neoplasia found anal cytology to have a sensitivity and specificity of 71% (95% CI, 61%–79%) and 73% (95% CI, 66%–79%), respectively, for detecting anal HSIL/cancer. These figures, they argue, are comparable to those described for cervical cytology with sensitivity and specificity which ranges from 52%–94% and 73%–97%, respectively [110]. Unfortunately they found poor concordance between cytological and histological grades (κ = 0.147) [110]; this has been replicated in numerous studies, most recently in a systematic review which only identified 2 out of 39 studies meeting the IANS HRA guideline recommendation that HSIL should be identified in more the 90% of cases with cytologic HSIL, most studies were below 75% [109]. This variability in positive predictive value of HSIL cytology has also been seen for cervical precancer [109]. 

There are some studies looking at carrying out anal hrHPV testing as an adjunct to cytology to improve its sensitivity and specificity. In MSM living with HIV this has been argued to be of little value given that more than 90% of this patient cohort are positive for anal hrHPV [108], moreover HPV testing has been shown to have high sensitivity but low specificity [111]; with hrHPV testing summary sensitivity and specificity estimates of 92% and 42% [109] and co-testing (cytology + HPV) sensitivity of 93% and 33% [109]. With that said anal hrHPV positivity in women with anogenital neoplasia is lower, with a systematic review reporting positivity ranging between 23% to 86% [112] and another review of 59% (n = 6) [109]. Wohlmuth et al. [111] carried out a study using cytology based screening for women over the age of 40 years with a history of cervical HSIL. They found 30.3% (96/317) of these women to have abnormal anal cytology and 31.9% (101/317) to be positive for anal HPV DNA [111]. Although the study did not directly compare cytology and HPV testing as primary screening modalities, they did find the presence of high-risk anal HPV to be the strongest predictor of abnormal anal cytology [111]. Finally, in women, Clarke et al. found anal HPV testing sensitivity and specificity of 91.1% and 47.1%, with a 27% (95% CI, 15–43%, τ^2^ = 0.70) risk of identifying HSIL among HPV positives. Again, supporting its potential role as an anal HSIL/SCC screening adjunct to cytology for women with genital dysplasia. 

#### 9.1.3. High Resolution Anoscopy

High resolution anoscopy is currently considered the gold standard for anal HSIL detection and to allow targeted sampling of abnormal tissue. There has been a move away from the conventional anal mapping which is inaccurate as lesions not easily detectable to the naked eye are often missed, this is especially true for intra-anal pathology. HRA utilises magnification from a conventional colposcope to examine the perianal region and squamocolumnar junction and as part of the procedure abnormal lesions are made more visible with the use of 5% acetic acid [113]. Anal HSIL is often aceto-white with mosaic-like-features or punctation which represent vascular changes. Lugol’s iodine can also be applied following acetic acid on lesions at the squamocolumnar junction, it can increase diagnostic accuracy as HSIL lacks Lugol’s uptake [113]. Suspicious lesions can then be biopsied. The procedure is normally carried out in clinic and well tolerated by patients, although women appeared to report higher pain scores [114]. The authors did comment that this may have been a result of the fact that many of the women in their study were undergoing multizonal assessment which included examination of the cervix, vagina, vulva, and anus. Despite this most of the female patients were satisfied with the care they had received and would have been happy to undergo the examination again [114]. This is important as control of disease is often not achieved in one session and most patients require, repeat examinations, biopsies, treatment and follow-up before disease is controlled [115]. 

HRA has been reported to have sensitivity and specificity ranging from 59% to 100% and 71% to 74%, respectively [116] with one study focusing on women with cervical neoplasia reporting sensitivity and specificity of 57.6% and 86.1% [116]. 

The main issue with HRA is that its diagnostic accuracy is very much user dependent, as reflected by the published variance in sensitivity and specificity. IANS has created recommendations and quality criteria [117] in order to standardise practice and set a standard for training and as a society is trying to drive HRA training; currently there are few trained clinicians and centres which is a limitation of this technique. 

### 9.2. Treatment

The ANCHOR trial [12,13] has provided the evidence to support the treatment of anal HSIL in the prevention of anal cancer. However, we have yet to determine which treatment modality is superior and this will also likely be influenced by the size of the lesion, the location of the lesion (intra-anal vs. perianal) and patient factors (immunocompromise, previous surgery, anatomical variation). Currently, treatment options can be divided in 2 categories: topical and ablative [118]. A third option is surgical excision, however it has high associated morbidity including the development of anal stenosis, faecal incontinence and disease persistence (50% in one study) [119]. There is simply insufficient research looking at the effectiveness of different anal HSIL treatments in the prevention of anal SCC, with significant heterogeneity in the response rates for all modalities [118]. Moreover, most published work has focused on PLWH with almost no research being carried out in women with anogenital dysplasia. Women with multizonal anogenital disease are unable to clear hrHPV and are at high-risk of recurrent disease [50]. They will likely require close surveillance and repetitive treatments with different modalities to achieve disease clearance, however there is a need for more research to be carried out in this patient group. 

### 9.3. Screening and Surveillance Pathways

There are no globally accepted recommendations for anal screening. There are also no guidelines advising on how to best manage women at high-risk of anal HSIL and SCC. A systematic review comparing available clinical guidelines for the management of anal SIL highlights that most guidelines name anal cytology as the screening tool of preference, despite its limitations in diagnostic accuracy [63]. Moreover, most also advise referral for HRA if abnormal cytology is identified [63]. With respect to treatment the review found little concordance between guidelines, some arguing anal HSIL should be treated and others advocating close surveillance instead [63]. This is not unexpected given that the review was published before the publication of the ANCHOR trial results [12,13]. As for follow-up and surveillance most guidelines recommended 3–6 monthly follow-up for 3–5 years with a combination of cytology +/−HRA, with one also recommending yearly HRA follow-up. 

Wohlmuth et al. [111] carried out a study in which women over the age of 40 with a history of cervical SIL or cancer were screened for anal dysplasia using anal cytology with HPV-DNA testing. All patients with abnormal anal cytology were then referred for HRA. If HRA was normal, they would get a repeat HRA assessment only if there was high-grade cytology, otherwise they would be sent for routine follow-up care. If HRA identified dysplasia then the patients would be offered treatment, although the scope of the study was not to assess treatment of anal HSIL. They had a total of 317 women in their study, 96 with abnormal cytology. Of these 96 patients 30 (31.3%) were found to have anal SIL on HRA targeted biopsy (9.5% of the total patient cohort), with the majority (20/96, 66.7%) having anal HSIL [111]. Although the aim of the study was not to propose an optimal screening approach, what it demonstrates is that anal HSIL screening and surveillance could at least be feasible in this patient population with the currently available detection tools. More importantly >90% of patients completed the study protocol, which arguably reflects the importance that patients see in detecting early HPV-related disease. 

Interestingly, there are a few studies investigating the cost and benefits of screening women with previous cervical dysplasia, which have promising conclusions. 

Cromwell et al. [120] looked at a model which integrated anal cytology screening with the cervical screening programme for women with cervical HSIL. They modelled their screening to continue for 20 years post cervical HSIL diagnosis in this patient subgroup and found that adding anal SIL screening to a pre-existing screening programme could in fact be cost- effective and beneficial for this patient cohort, although it may not contribute to a meaningfully different quality-adjusted survival.

Ehrenpreis et al. [121] constructed a dynamic model looking into the cost-effectiveness and health benefits of both screening for anal HPV as well as treatment of anal HSIL in patients with cervical cancer; they found these to be cost-effective and important in the prevention of anal cancer and reduction in anal cancer deaths. Their results were based on an assumed 98% cure rate of anal HSIL with electrocautery, however sensitivity analysis demonstrated the benefits to occur even with cure rates as low as 38%. Costs in the screened group included initial testing for anal HPV, annual anal cytology, the cost of treatment for anal HSIL and annual follow-up for patients treated with HSIL (including HRA and biopsy). 

Given limited HRA availability, a hypothetical screening programme for high-risk women with a history of persistent cervical hrHPV, HSIL or SCC but no known anal hrHPV or HSIL could involve the initiation of yearly anal cytology and HPV testing starting at the age of 40 years; given that patients with cervical HSIL over this age have been shown to have a higher incidence of anal cancer [4] and that anal cancer is most common over the age of 50 years [2,10]. This could be integrated in the cervical screening programme or take place in the shape of a self-sampling anal swab campaign, which has been shown be a viable model for anal and cervical sampling [122]. Women with positive cytology could then be referred for HRA, see Figure 2. Those with anal HSIL could then receive treatment for their condition and enter an anal HSIL surveillance pathway, which could comprise: 3 monthly HRA until disease free, followed by 6 monthly HRA for a year, yearly HRA for 5 years and then entry back into normal screening. Those negative for HSIL could re-enter normal screening. Equally, women with 2× consecutively positive anal hrHPV swabs could be referred for HRA and then re-enter normal screening or the HSIL treatment and surveillance pathway depending on the findings. 

### 9.4. World Health Organisation (WHO) Screening Criteria for the Secondary Prevention of Anal Cancer

The current available evidence does not yet satisfy all the WHO screening secondary prevention criteria [123], see Figure 3. It is clear that anal SCC is an important health problem of which prognosis is strongly related to stage at presentation [11], there is an adequate understanding of the natural history of anal HSIL and SCC (see Section 2 and Section 3), anal HSIL represents a recognizable latent or early symptomatic stage (see Section 3) and there are suitable tests and examinations available to facilitate the detection of anal HSIL (see Section 9.1). In terms of acceptability of tests, the current evidence suggests that even HRA which is arguably the most invasive of the detection tools is relatively well accepted [114,124,125], although women have been shown to report higher pain scores [114]. Moreover, acceptability will vary with patient population, anal history, and cancer concern. There is currently no official accepted treatment for patients with recognized anal HSIL; the ANCHOR trial [12,13] provided evidence supporting the treatment of anal HSIL but did not determine the best treatment modality, which as discussed in Section 9.2, will likely vary between different high-risk patient groups. More importantly, research looking at the effectiveness of different treatment modalities in women with genital dysplasia is lacking. There is of course an accepted treatment pathway for patients with anal cancer, consisting in local resection, chemoradiotherapy and salvage abdominoperineal resection, depending on the staging of disease [96,98,101,105], however the aim of screening would be to detect and treat disease before it is invasive. Although there is a good understanding around which patient groups are at highest risk of anal HSIL and SCC [4], there is still no agreed policy on whom to treat as patients; there is a need to establish an incidence level above which screening is favourable. More work determining the cost-effectiveness needs to be carried out; for high-risk women, only hypothetical models are currently available in the literature [120,121]. Lastly, both criteria 9 and 10 (see Figure 3) will depend on the availability of resources and expertise which vary significantly between different geographical areas, especially when considering the availability of HRA.

## 10. The Role of the HPV Vaccination

The role of the HPV vaccine in the primary prevention of cervical dysplasia is unquestionable. A recent publication by Falcaro et al. [126] has shown the introduction of the HPV immunisation program in England in 2008 to reduce the risk of developing cervical HSIL by 97% in those patients vaccinated at the age of 12–13 years, almost eliminating cervical cancer in women born after the 1 September 1995. The impact this will have on other genital and anal dysplasia remains to be seen, however it is likely going to drastically decrease the incidence all anogenital HPV driven pathology. 

The role of the vaccine in the secondary prevention of anogenital HPV driven dysplasia is less understood. There are a number of observational studies which report therapeutic effects in patients already infected by HPV; one study using the quadrivalent vaccine demonstrating a 50% risk reduction in cervical disease recurrence in the vaccinated group after surgical treatment of cervical dysplasia [127]. Another study also using the quadrivalent vaccine in men with anal HSIL, showing a 50% reduction in anal HSIL recurrence in the vaccinated group [128]. A systematic review looking into the role of the vaccine on HPV infection and recurrence of HPV related disease after local surgical treatment of genital disease concluded that the vaccination might reduce the risk of cervical dysplasia recurrence after treatment but evidence was lacking for the recurrence of vulval, vaginal and anal dysplasia [129]. In contrast, a randomised control trial assessing the effectiveness of the quadrivalent vaccine in the prevention of new HPV infection or reduction in anal HSIL recurrence in PLWH was terminated early for futility, with a reported vaccine efficacy of only 22% for the prevention of persistent infection after 3.4 years [130]. 

More research in its role in the secondary prevention of anogenital dysplasia in women with a background of genital pathology is required. The VIVA trial, looking to test whether Gardasil-9 can reduce the risk of HSIL recurrence by 50% in previously unvaccinated individuals recently treated for anal or vulval HSIL [131], will hopefully be able to answer some of these questions. 

## 11. Conclusions

With no standardised, evidence based, anal HSIL management guidelines for any high-risk patient group and very limited research focusing on HIV negative high-risk women we are clearly still far from being able to establish an effective anal HSIL screening, treatment, and surveillance programme for this patient cohort. With that said, the ANCHOR [12,13] trial has significantly impacted how we look at anal HSIL, highlighting the importance of early identification and treatment of HSIL in the prevention of SCC and demonstrating that even active surveillance is likely inadequate in high-risk populations. 

We need more research to focus on women with genital dysplasia as they represent a high-risk population for anal HSIL and SCC, which accounts for a significant proportion of women developing anal cancer. With increasing numbers of women presenting with advanced disease every year, we have a duty to start educating these women about their anal cancer risk, by raising awareness and establishing services to facilitate the early detection and treatment of anal HSIL before it progresses to cancer. More evidence is required to fully comply with WHO criteria with respect to anal HSIL screening in high-risk populations, especially in women with hrHPV driven genital pathology, however, given that such services are available for PLWH with a similar risk profile it is perhaps worth trialing this in this population too.

## Figures and Tables

**Figure 1 cancers-15-00060-f001:**
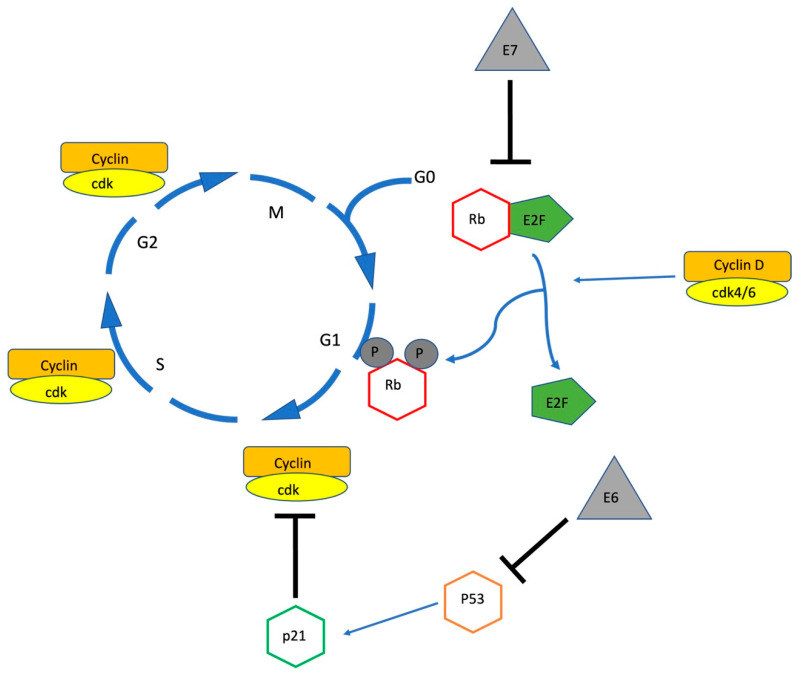
CyclinD/cdk4/6 complex phosphorylate pRB causing it to release E2F and allow progression of the cell cycle past the G1/S Restriction Point. Oncoprotein E7 acts by inhibiting pRb. Among other functions p53 activates p21, a cdk inhibitor regulating the G1/S checkpoint. Oncoprotein E6 promotes the degradation of p53.

**Figure 2 cancers-15-00060-f002:**
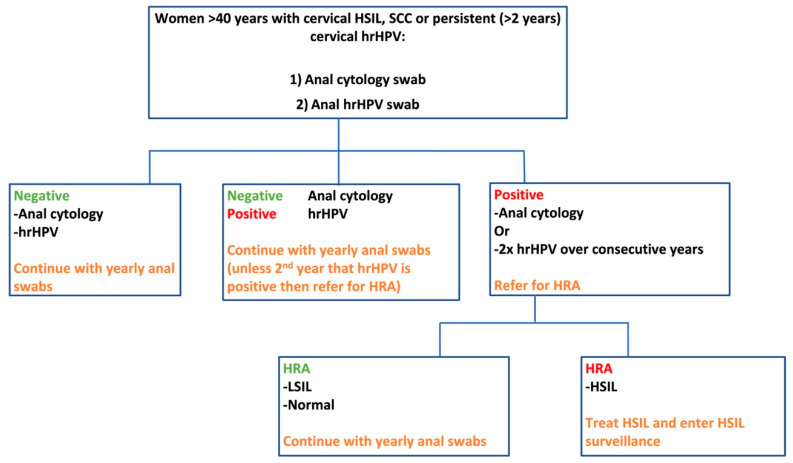
Hypothetical anal cancer screening model for the early detection and treatment of anal HSIL.

**Figure 3 cancers-15-00060-f003:**
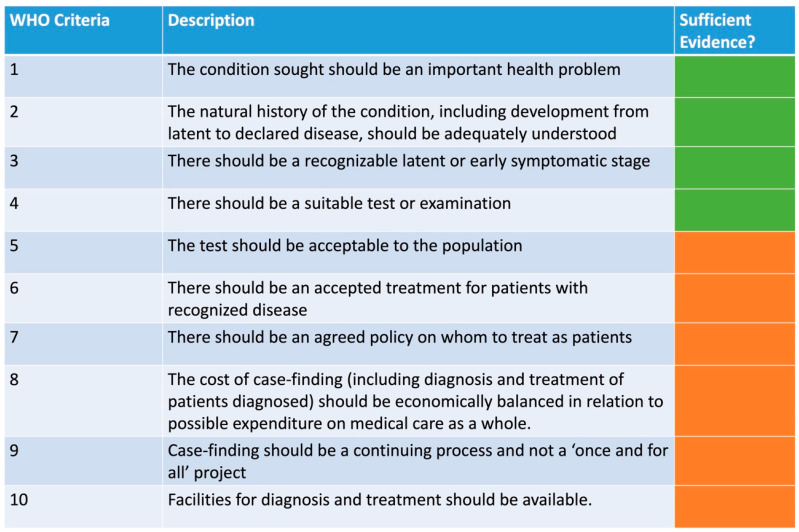
WHO screening criteria. Key: Green = good evidence, Orange = moderate/circumstantial evidence.

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
