# Peer review of "Anal Cancer in High-Risk Women: The Lost Tribe"

_cancers, 2022, doi:10.3390/cancers15010060_

Round 1

Reviewer 1 Report

The manuscript of Lupi et al. is a review of the current knowledge on anal cancer with a focus on clinical strategies to reduce cancer incidence and mortality in women. The manuscript is very well written, and the topic is very important. The manuscript could be improved as follows:

 Minor issues

 HPV integration is discussed in the introduction, but HPV16, the cause of most anal cancer does not integrate into about 1/3 of cervical tumors. Is there comparable data on anal cancers?

 Figure 2 is somewhat confusing, and I appreciate there is incomplete data.

If 24% of HSIL regress to LSIL and there is a 10% lifetime risk of progression to SCC, do 66% remain HSIL for life?

 l.135   10,00 should be 10,000

 l167-171. These 2 paragraphs could be combined.

L321-2 Similarly, this one-sentence paragraph can be combined with the following paragraph. Same with l.539-40

l. 377 ‘The HPV vaccination’ … should be ‘HPV vaccination…’

l. 495 ‘especially if intra-anal.’ should be ‘especially for intra-anal’ ? or the period removed?

l. 504 ‘their women cohort’ would read better as ‘the women in this study’

 The text in Figure 3 is hard to read, and the font should be increased to be much larger. Same for Figure 4.

Reviewer 2 Report

Anal squamous cell carcinoma (SCC) is a relatively rare cancer, but is surprisingly highest in women compared to men. More importantly, women have been shown to be more likely to present late with advanced cancers.

The authors of this manuscript have sufficiently elaborated and critically reviewed the most relevant literature in their chosen topics, i.e., anal cancer in high-risk women with a previous history of genital high-grade squamous epithelial lesions (HSIL) and SCC. Specifically, the manuscript summarized high-risk HPV (hrHPV) pathogenesis, anal squamous intraepithelial lesions and their natural history, and risk factors for anal hrHPV colonization in women. They also thoroughly discussed various factors (smoking, immunosuppression, HIV) for chronicity of hrHPV infection in high risk women. Moreover, they critically reviewed the current literature that have reported screening, surveillance and treatment of anal HSIL in high-risk women.

Overall, it's an excellent comprehensive review, with up-to-date information and of significance to the field. The manuscript is also very well written.
